METHODS AND PROTOCOLS

Novel Systems Biology Techniques

# phyloFlash: Rapid Small-Subunit rRNA Profiling and Targeted Assembly from Metagenomes

Harald R. Gruber-Vodicka,ᵃ Brandon K. B. Seah,ᵃ* Elmar Pruesseᵇ

ᵃMax Planck Institute for Marine Microbiology, Bremen, Germany

ᵇDepartment of Medicine, Division of Biomedical Informatics and Personalized Medicine, University of Colorado Denver Anschutz Medical Campus, Aurora, Colorado, USA

Harald R. Gruber-Vodicka and Brandon K. B. Seah contributed equally. Author order was determined alphabetically by family name.

**ABSTRACT** The small-subunit rRNA (SSU rRNA) gene is the key marker in molecular ecology for all domains of life, but it is largely absent from metagenome-assembled genomes that often are the only resource available for environmental microbes. Here, we present phyloFlash, a pipeline to overcome this gap with rapid, SSU rRNA-centered taxonomic classification, targeted assembly, and graph-based binning of full metagenomic assemblies. We show that a cleanup of artifacts is pivotal even with a curated reference database. With such a filtered database, the general-purpose mapper BBmap extracts SSU rRNA reads five times faster than the rRNA-specialized tool SortMeRNA with similar sensitivity and higher selectivity on simulated metagenomes. Reference-based targeted assemblers yielded either highly fragmented assemblies or high levels of chimerism, so we employ the general-purpose genomic assembler SPAdes. Our optimized implementation is independent of reference database composition and has satisfactory levels of chimera formation. phyloFlash quickly processes Illumina (meta)genomic data, is straightforward to use, even as part of high-throughput quality control, and has user-friendly output reports. The software is available at https://github.com/HRGV/phyloFlash (GPL3 license) and is documented with an online manual.

**IMPORTANCE** To track organisms across all domains of life, the SSU rRNA gene is the gold standard. Many environmental microbes are known only from high-throughput sequence data, but the SSU rRNA gene, the key to visualization by molecular probes and link to existing literature, is often missing from metagenome-assembled genomes (MAGs). The easy-to-use phyloFlash software suite tackles this gap with rapid, SSU rRNA-centered taxonomic classification, targeted assembly, and graph-based linking to MAGs. Starting from a cleaned reference database, phyloFlash profiles the taxonomic diversity and assembles the sorted SSU rRNA reads. The phyloFlash design is domain agnostic and covers eukaryotes, archaea, and bacteria alike. phyloFlash also provides utilities to visualize multisample comparisons and to integrate the recovered SSU rRNAs in a metagenomics workflow by linking them to MAGs using assembly graph parsing.

**KEYWORDS** SSU, gene assembly, metagenomics, taxonomic profiling

Shotgun metagenomics is a powerful tool to explore the functions of microbial communities and to determine their phylogenetic or taxonomic composition (1, 2). It can avoid biases and artifacts associated with PCR-based amplicon methods, such as primer bias (3) and sequence chimerism (4, 5). Shotgun sequencing also yields more information because it samples from whole genomes rather than a single marker. Various approaches have been applied to profile the composition of shotgun metagenomic libraries. A common choice is local alignment against reference sequences,

Address correspondence to Harald R. Gruber-Vodicka, hgruber@mpi-bremen.de.

* Present address: Brandon K. B. Seah, Max Planck Institute for Developmental Biology, Tübingen, Germany.

Start the full-cycle rRNA approach from metagenomes with phyloFlash. Diversity assessment + SSU reconstruction from specifically curated dB. Get full length SSUs, compare samples and more. Bac, Arch and Euk welcome. #phyloFlash #mSystems

which may target a specific region in a single gene (6, 7), curated sets of conserved single-copy genes (8, 9), or whole-genome data (10, 11). Others use alignment-free methods such as k-mer matching to classify reads (12–14), and some tools also go beyond classification to perform targeted assembly of specific genes (15, 16). Regardless of the method that is used for taxonomic or functional profiling, the results will be limited by the reference data available. For example, target organisms or their close relatives may not yet be represented in the database, and horizontal gene transfer can result in conflicting phylogenetic signal, especially in prokaryotic microbes (17, 18).

In molecular ecology, the gene for the small-subunit rRNA (SSU rRNA) is the most important marker because it can be readily used to link sequences to actual cells. The SSU rRNA is available in high copy numbers in ribosomes and can be accessed through the well-established molecular probing technique of fluorescence *in situ* hybridization (FISH) (19, 20). Its value in imaging-based analyses together with its high phylogenetic reliability due to low rates of horizontal transfer has made the SSU rRNA gene the best-sampled marker in terms of phylogenetic diversity (21). Even with the current advances of metagenomics, where drafts of microbial genomes can be automatically extracted ("binned") from metagenome assemblies (22), the SSU rRNA gene has the same essential role and is crucial for phylogenetics, imaging, and experimental verification. However, despite all the progress on automated binning (23), most metagenome-assembled genomes (MAGs) do not contain even fragments of the SSU rRNA gene, not to mention the full gene (1, 24). Another potential drawback of the SSU rRNA gene as a marker for molecular ecology is that a single genome may have multiple copies of the rRNA gene operon, in both eukaryotes and prokaryotes (25), so the abundance of rRNA sequences may not directly reflect cellular abundance in a community.

Ideally, we would like to leverage the vast existing knowledge base of the SSU rRNA gene in (meta)genomics projects for several different outcomes: taxonomic profiling without assembly (6, 7), targeted assembly of full-length sequences for phylogenetics and probe design (15, 16, 26, 27), and linking SSU rRNA sequences to complete genomes (28). For each of these aims, separate software tools have already been developed, each with their own merits and shortcomings (16). However, these should be considered together because they all involve the same target gene, and improvements to each task can directly lead to improvements in the others.

Here, we describe phyloFlash, an open-source pipeline for the rapid profiling and targeted assembly of SSU rRNA from metagenomes (Fig. 1). We use the pipeline to evaluate the trade-offs in different steps of taxonomic profiling and targeted assembly, starting from the preparation of the reference database, to the extraction and assembly of target reads, and finally to linking the assembled sequences to genome bins. We compare the performance of rRNA-specialized versus general-purpose read mappers and assemblers, by using both simulated read data and real-world environmental metagenomes. We also reanalyze published data to show how the pipeline can be used to assess and compare the composition of metagenomes. Finally, we show how genome bins from metagenomes can be linked to SSU rRNA sequences, to bridge the divide between genome-centric metagenomics and experimental molecular ecology.

## RESULTS AND DISCUSSION

**Reference database artifacts influence targeted read extraction.** The preparation of the reference database affects the number and quality of reads extracted for downstream analysis, even when using a curated resource such as the SILVA databases. Although potential contaminants and low-complexity sequences represented only a small fraction of the total sequences, they had a disproportionate effect on the reads recovered. In release 132 of the SILVA SSU Ref NR99 database (total 695,171 sequences, 1.006 Gbp), we detected 42.88 kbp (in 151 sequences) of matches to partial large-subunit rRNA (LSU rRNA) and 1.102 Mbp (in 56,075 sequences) of potential vector contamination. Furthermore, sequence repeat and low-complexity regions constituted 23.31 kbp.

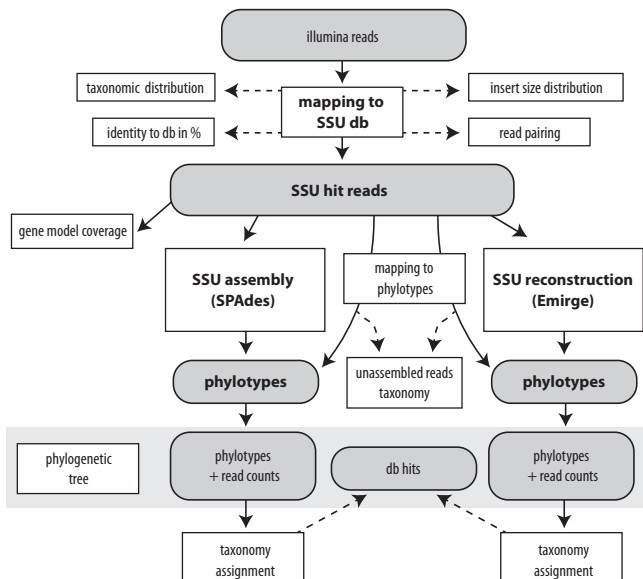

**FIG 1** Flowchart of the phyloFlash pipeline. Gray boxes depict main data files, and white boxes with bold print show processes. Solid black arrows indicate primary workflow; dashed lines indicate processing of secondary output.

Fewer reads were extracted with a filtered database, but their quality was improved. For example, in the library with NCBI:SRA accession no. ERR315856, 32,937 reads were recovered by SortMeRNA (E-value of $10^{-5}$) before filtering, but only 17,227 after filtering. Moreover, the mean informational redundancy decreased from 0.0748 to 0.0151, indicating that fewer low-complexity sequences were extracted (Fig. 2). The degree of impact from filtering depends on the metagenome library composition, but similar results were observed when reads were extracted with BBmap, and for other metagenome libraries (see Fig. S1 in the supplemental material). Subsequent results reported here used the filtered database.

Some SSU rRNA genes may indeed contain low-complexity regions; nonetheless, it is still desirable to remove them from the reference database. This is because any low-complexity sequence in the library (e.g., from eukaryotic genomes), regardless of origin, will tend to map to these regions. This would skew any analysis of the taxonomic composition of a metagenome that is based on mapping to reference sequences, and it could also affect assembly and other downstream analyses.

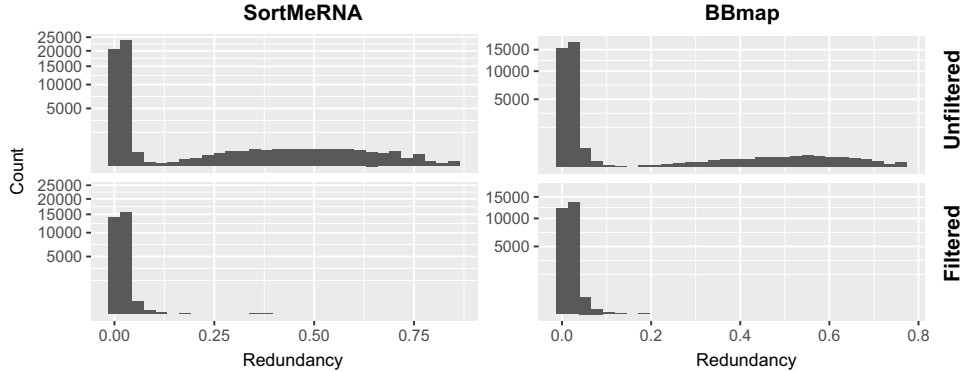

**FIG 2** Database filtering removes low-complexity read hits. Informational redundancy (k = 5) in reads extracted by SortMeRNA (left) or BBmap (right) using unfiltered (above) and filtered (below) reference databases, for library with NCBI:SRA accession no. ERR315856.

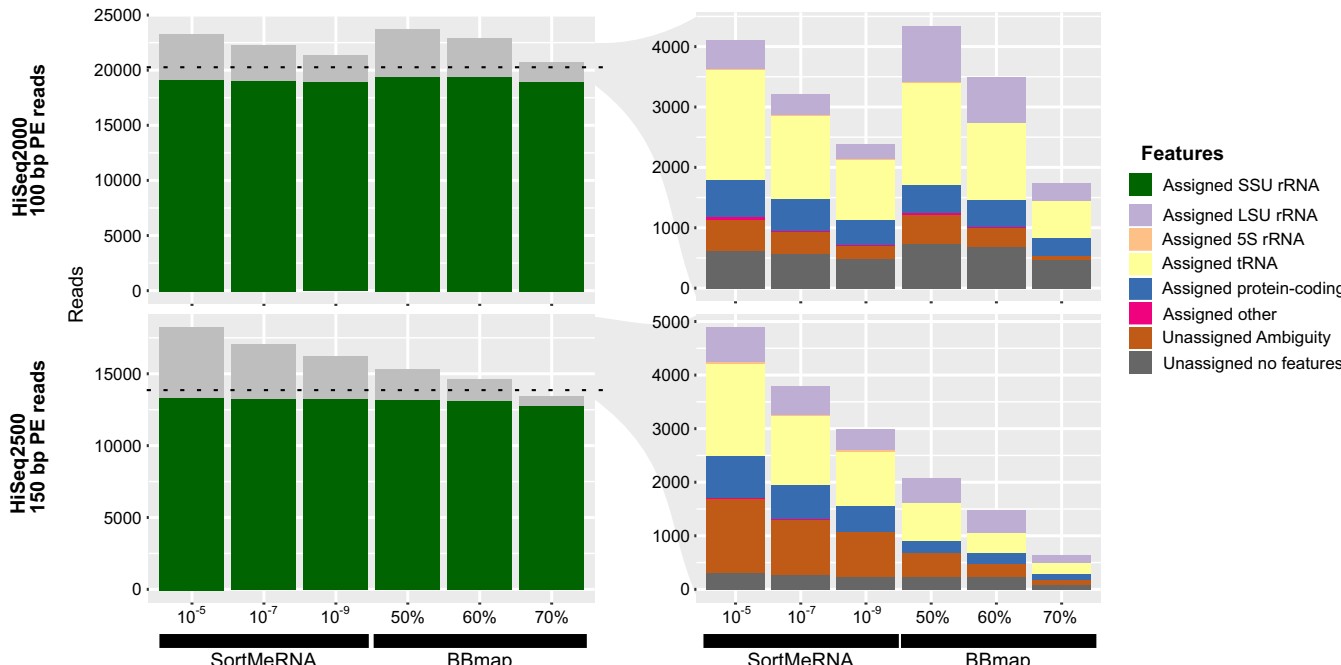

**FIG 3** False-positive read extraction depends on stringency cutoffs, but only BBmap results improve with read length. Origin of reads extracted by SortMeRNA and BBmap at different parameters (horizontal axes) from simulated metagenome of 100-bp paired-end (PE) reads (above) or 150-bp PE reads (below), showing the proportion of total originating from actual SSU rRNA features (left) and the origin of non-SSU rRNA-extracted reads (right). Horizontal lines indicate the total actual SSU rRNA origin reads in each library (20,213 and 13,829, respectively).

**BBmap provides similar sensitivity and higher selectivity versus SortMeRNA at fivefold faster runtimes.** Different tools have been developed for extracting SSU rRNA reads from shotgun metagenomes. Those that depend on alignment to a reference database ("mapping") can be used to derive an approximate taxonomic assignment from the closest-matching reference sequences, which is not possible with model-based methods (e.g., hidden Markov models [HMMs] or covariance models). We compared two reference-based tools for read extraction: SortMeRNA, originally designed to identify rRNA reads in metatranscriptomes (29), and BBmap, a general-purpose mapper.

The performance of these read extraction tools was dependent on their stringency settings (E-value cutoff for SortMeRNA, minimum mapping identity for BBmap), as well as properties of the read library such as read length, insert size (for paired-end reads), and error profile. We simulated two Illumina paired-end metagenomes from the same 10 RefSeq bacterial genomes ("set 1" in Data Set S1 in the supplemental material): a HiSeq2000 library with 100-bp reads ($220 \pm 110$-bp insert size [mean $\pm$ standard deviation]) and a HiSeq2500 library with 150-bp reads ($300 \pm 110$-bp insert size). In terms of their sensitivity, i.e., the fraction of reads originating from annotated SSU rRNA features that they could recover, both tools had a similar performance (ca. 94 to 96%, depending on settings) with 100-bp reads, and SortMeRNA performed slightly better (96 to 97%) than BBmap (93 to 96%) with 150-bp reads (Fig. 3; see also Table S1 in the supplemental material). However, in terms of selectivity, i.e., the proportion of true SSU rRNA reads among the total reads recovered, BBmap performed better with 150-bp reads than 100-bp reads (86 to 95% versus 82 to 92%), whereas SortMeRNA performed worse (73 to 82% versus 82 to 89%). The main source of false-positive hits was tRNA genes (Fig. 3). BBmap was also about fivefold faster than SortMeRNA when run on the same machine (Table S1).

**SortMeRNA can yield more assembled reads than BBmap with environmental metagenomes.** In real metagenomic libraries, the origin of each read is not known *a priori*. The extraction of SSU rRNA reads can be evaluated only indirectly, by comparing the overlap in reads recovered by both tools. An alternative measure for the quality of

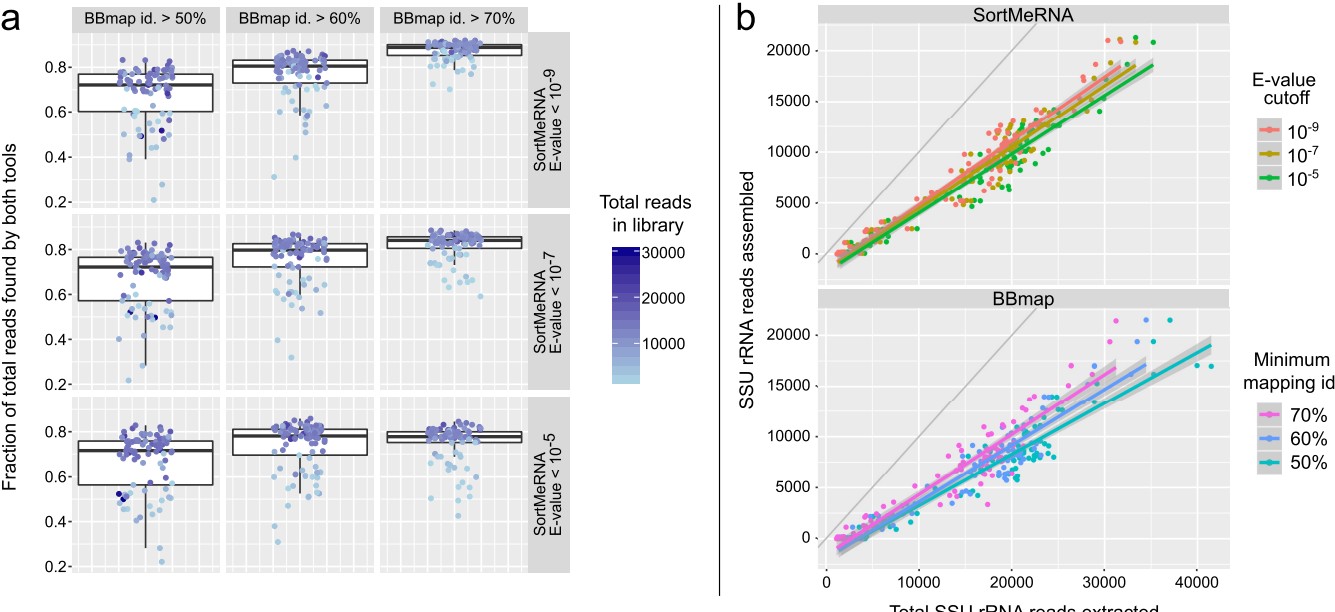

**FIG 4** Mapper and cutoff settings influence SSU rRNA read extraction from Tara Oceans metagenome libraries but yield highly similar assembly ratios. (a) Fraction of reads extracted by both tools at different BBmap minimum identities (id.) (horizontal) and SortMeRNA E-value cutoffs (vertical). (b) Reads assembled by SPAdes versus total reads extracted for different parameter settings (colors) of SortMeRNA (top) and BBmap (bottom), overlaid with linear regression lines and 1:1 line for reference (gray).

read extraction is the fraction of extracted reads that can be assembled to full-length gene sequences. We performed both comparisons with 85 environmental metagenomes from the surface ocean seawater habitat (Data Set S1).

More restrictive settings (lower E-value cutoff for SortMeRNA, higher minimum identity for BBmap) resulted in more overlap between the reads recovered by both tools, with a mean 87.2% overlap for the most restrictive settings tested (SortMeRNA E-value of $10^{-9}$, BBmap 70% minimum identity) versus a mean 66.6% overlap for the least restrictive settings tested ($10^{-5}$, 50%) (Fig. 4). More permissive settings for a given tool resulted in more reads extracted only by that tool (Fig. S2). Similarly, more restrictive settings resulted in a higher fraction of reads that could be assembled into full-length SSU rRNA sequences, e.g., for SortMeRNA, the mean fraction was 45.4% at an E-value cutoff of $10^{-9}$ versus 40.6% at an E-value of $10^{-5}$, and for BBmap, mean 41.1% at minimum mapping identity 70% versus 34.2% at >50% minimum identity. This can be seen in Fig. 4b as the slope of the relationship between the number of assembled reads versus the total extracted reads increased as settings became more restrictive, irrespective of the extraction tool.

The absolute number of reads that can be assembled does not vary much for most libraries with the cutoff settings, and on average, SortMeRNA performs better than BBmap in extracting reads that can be assembled, with this improvement being more pronounced for some libraries (Fig. S3). The mean ratio of assembled reads extracted by BBmap to SortMeRNA is between 85.9% and 90.6% depending on the program settings being compared (Table S2), but for more than half the libraries, both BBmap and SortMeRNA have comparable performance: out of the 81 libraries with a nonzero number of assembled reads, the BBmap/SortMeRNA ratio was >90% for 47 libraries, and it was >95% for 37 libraries.

We chose BBmap as the default read extraction tool as it was much faster in our tests with both simulated and real metagenomes and had similar performance in most cases. For more detailed but time-intensive analyses and to investigate the possibility of highly divergent taxa by targeted assembly, both SortMeRNA and more permissive settings for both tools can be tested. For BBmap, this comes with the caveat that the taxonomic summary would likely become less reliable. More permissive settings for

**TABLE 1** Comparison of targeted assembler software for SSU rRNA sequences

| Tool | Read extraction | Database | Method | Taxon classifier | Availability | Documentation | License | Last release (yr-mo-day)[a] |
|---|---|---|---|---|---|---|---|---|
| Emirge | Bowtie (v1) | SILVA | Expectation maximization | (From mapping) | GitHub, Conda | Readme, user forum | GPL3 | v0.61.1 2016-12-03 |
| REAGO | Infernal (CM) | NA | Overlap graph, pruning, paired-end guided path-finding | NA | GitHub, Conda | Readme | BSD | v1.1 2015-12-18 |
| Matam | SortMeRNA | SILVA NR95 | Overlap graph, compression, SGA assembler on subgraphs | RDP classifier | GitHub, Conda, Docker | Readme | AGPL3 | v1.6.0 2019-10-09 |
| RAMBL | Bowtie2 | Greengenes | Taxonomic tree search and Dirichlet process clustering | RDP classifier | GitHub | Readme | Not specified | NA last commit 2017-03-18 |

[a]As of 1 October 2020. NA, not available.

initial read extraction may increase the probability of false-positive hits but can also make it possible to detect possible distant relatives. Given the popularity of SSU rRNA as a phylogenetic marker and the fact that SSU rRNA sequences in public databases represent the accumulated results of several decades of taxon sampling, the more restrictive settings should be adequate for the routine assessment of taxonomic composition using read mapping hits to reference sequences. Defaults of E-value $< 10^{-9}$ for SortMeRNA and minimum identity of $>70\%$ for BBmap were therefore chosen for the phyloFlash pipeline.

**General-purpose assembler SPAdes yields longer and more divergent scaffolds at acceptable levels of chimerism.** Existing software tools that have been developed for the targeted assembly of SSU rRNA sequences employ various algorithms and reference data (Table 1). The ideal assembler should have the following properties: be able to assemble full-length gene sequences, while avoiding chimeric assemblies, be able to assemble highly divergent sequences, and yet differentiate between closely related strains or paralogs. In reality, optimizing for one metric may preclude another. For example, attempting to recover strain-level variation will result in more fragmented assemblies, because insert sizes from Illumina shotgun libraries are usually too short to resolve full-length sequences that differ only in the variable regions of the SSU rRNA gene.

We compared three different assemblers, two targeted-assembly tools Emirge (15) and Matam (16) and the general-purpose genome assembler SPAdes (30), on SSU rRNA reads extracted with SortMeRNA using the filtered SILVA 132 SSU Ref NR96 reference database and an E-value cutoff of $10^{-5}$. To ensure a uniform comparison, we used only sequences matching at least 60% of a full-length SSU rRNA HMM model.

We first tested the assemblers on the same simulated metagenome ("set 1") previously used to test read extraction, containing phylum-level diversity. This data set originally contained a total of 25 SSU rRNA gene copies from 10 species, with between 71.1 and 99.9% (mean 80.3%) nucleotide identity. These reduced to 11 sequences when clustered at 99% identity (Data Set S1). SPAdes assembled the same 10 scaffolds (one per genome) each time from both the 100- and 150-bp libraries, whereas the scaffolds assembled by Emirge and Matam differed between the two libraries (Fig. S4). Although Emirge also assembled 10 scaffolds from each library, some were chimeric sequences that did not have a close match to any of the original sequences. Scaffolds assembled by Matam were often fragmented, e.g., from the 100-bp library, full-length sequences (i.e., $>60\%$ of SSU rRNA gene) were not assembled for four genomes.

Some organisms have multiple SSU rRNA gene copies which may be only slightly divergent and pose a challenge to assembly. Included in the simulated metagenome

"set 1" was *Escherichia coli* K-12 MG1655 which has seven copies that fall into two clusters at 99% identity. SPAdes assembled only one version, whereas Matam assembled both (Emirge assembled none). However, Matam also assembled more chimeras and fragments that clustered with the genuine *E. coli* sequences, and the sequences differed between the 100-bp and 150-bp libraries (Fig. S4). This suggests that reference-based assemblers like Emirge and Matam are more sensitive to the input read length or insert size and that they may also produce spurious assemblies, especially with clusters of close sequences, and are hence less consistent.

We next tested the three assemblers on two simulated metagenomes containing strain-level diversity. "Set 2" comprised 25 genomes of *Bacteroides* spp. belonging to 25 strains in four species, with 25 SSU rRNA genes of between 94.6 and 99.9% nucleotide identity. "Set 3" was a more realistic mixture comprising 16 genomes of *Bacteroides* spp. belonging to 15 strains in 10 species, with SSU rRNA genes between 90.4 and 99.9% identity. Unexpectedly, we found unrelated SSU rRNA sequences in two of the *Bacteroides* genome assemblies in set 3 (GCA_003458755.1 and GCA_003435895.1 [Fig. S5B]), which may represent contaminants or errors in metagenome binning. This underlines the value of SSU rRNA-targeted analyses even when whole-genome data seem to be available. The unrelated sequences were retained for the read simulation, but are not counted in the discussion below.

Reference-based assemblers were generally expected to perform better than *de novo* assemblers with closely related sequences that differ only at the strain and species level, as simulated here. For set 2, the *de novo* assembler SPAdes failed to assemble any sequences from the 100-bp paired-end library and assembled two sequences with the 150-bp library, one clustering with *Bacteroides eggerthi* and the other a chimeric sequence (Fig. S5A). For set 3, SPAdes assembled sequences clustering with 6 of the 10 input *Bacteroides* spp. from the 150-bp library, but only one, chimeric, sequence from the 100-bp library (Fig. S5B). Emirge was more consistent in both set 2 and set 3. For set 2, Emirge consistently reconstructed three sequences from both the 100-bp and 150-bp libraries: one representative each for the *B. eggerthi* and *Bacteroides stercoris* clusters and one chimeric sequence, whereas for set 3, Emirge reconstructed sequences clustering with 8 of 10 *Bacteroides* spp. from both 100-bp and 150-bp libraries, in addition to a variable number of chimeric sequences (3 with 150-bp library, 6 with 100-bp library). In comparison, Matam reconstructed many more sequences from both set 2 (18 from 100-bp library, 20 from 150-bp library) and set 3 (29 from 100-bp library, 27 from 150-bp library). However, these sequences were much more fragmented: for set 2, only 6 (100-bp library) and 8 (150-bp library) were above the 60% full-length criterion, and for set 3, only 13 (100-bp library) and 11 (150-bp library) sequences were above this criterion.

Similar to the results with the "set 1" simulated metagenome, Matam tends to overpredict and produce many short, fragmented reconstructions, whereas SPAdes and Emirge tend to collapse diversity into fewer sequences. However, none of the assemblers produced a sequence that exactly corresponded to each of the input SSU rRNA sequences, demonstrating the inherent limitations of short-read sequencing when the inputs have high sequence identity, as in the case of strain-level diversity.

Finally, when the three assemblers were tested against the real-world environmental metagenomes, there was a strong linear relationship between the number of SSU rRNA reads extracted per library and the number of scaffolds (Emirge $R^2 = 0.683$; Matam $R^2 = 0.558$; SPAdes $R^2 = 0.669$; $P < 2 \times 10^{-16}$ for all [Fig. 5]). Emirge assembled the most scaffolds per library (mean of 16.9 for Emirge, 6.95 for Matam, 7.48 for SPAdes), but it performs poorly by other metrics. Emirge scaffolds have more ambiguous bases (Emirge mean 2.775 undetermined bases [Ns] per scaffold; other assemblers 0), and are more likely to be chimeric (mean uchime_ref score of 2.65 for Emirge, 0.137 for Matam, and 1.71 for SPAdes). Matam scaffolds have the lowest chimerism scores, but they are shorter (mean 1,193 bp) than those from Emirge (1,420 bp) or SPAdes (1,368 bp). Matam scaffolds also have on average high identity to reference sequences in the database (mean of 99.8% for Matam, 97.7% for Emirge, and 99.0% for SPAdes).

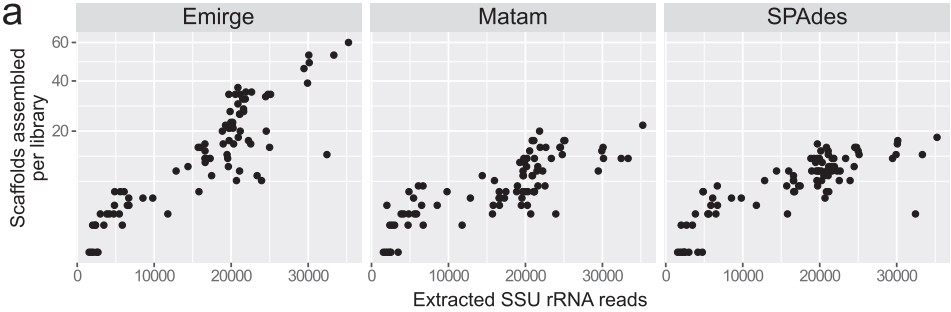

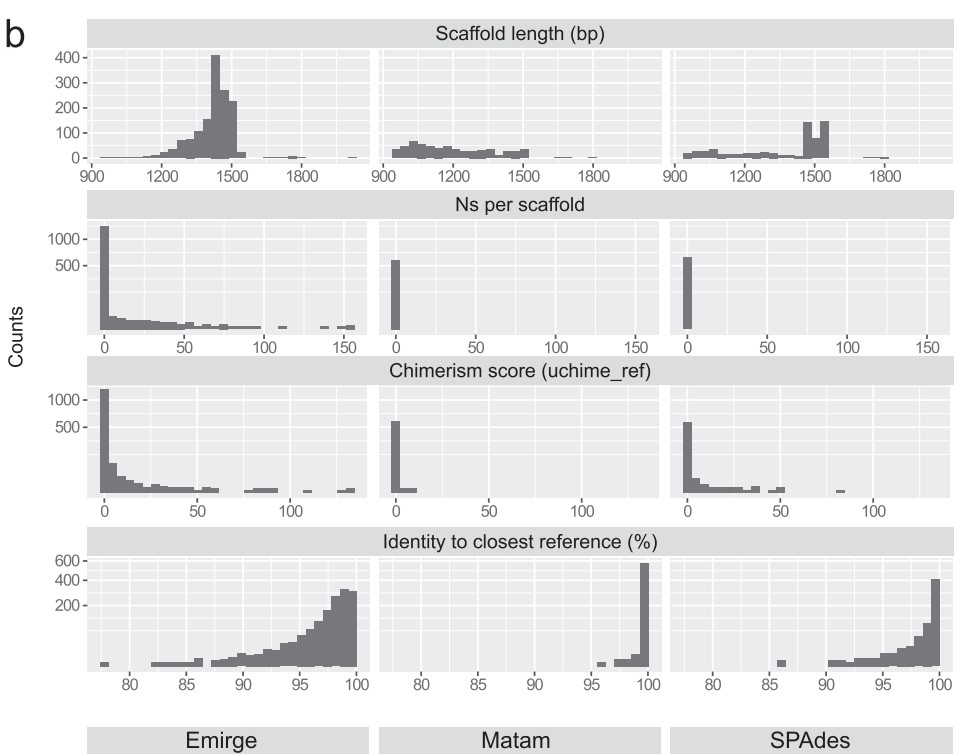

**FIG 5** Comparison of scaffolds assembled by SPAdes, Emirge, and Matam using reads extracted by SortMeRNA (E-value of $10^{-5}$, filtered reference database) from 85 Tara Oceans metagenomic libraries. (a) Plot of scaffolds assembled per library versus number of extracted SSU rRNA reads. (b) Histograms of per-scaffold metrics for each assembly tool: scaffold length (bp), undetermined bases (Ns) per scaffold, uchime_ref chimerism score, and identity to closest reference sequence.

Chimeric sequences are an acknowledged problem, and steps are taken to flag them in SILVA (31) but it is not possible to eliminate them entirely as the data ultimately comes from the wider scientific community. Chimera detection by reference-based methods, such as the uchime_ref score, are dependent on the reference database being free of chimeric sequences and having sufficiently close relatives to the query sequence. However, this cannot be guaranteed for public databases, where the deposited sequences have been assembled by different methods and tools, including reference-based assemblers that are known to produce chimeras, such as Emirge (16). Once chimeric sequences have been established in the reference database, subsequent chimerism scoring becomes less reliable. Based on our observed uchime_ref scores, some of the sequences assembled by either Emirge or SPADes may turn out to be chimeric, but in each case they will likely need to be examined separately.

Although Matam appears to be optimized for minimizing chimerism, this is at the expense of assembling more fragmentary scaffolds, and possibly overlooking SSU rRNA sequences that do not have a close relative in the reference database. With closely

related sequences (i.e., strain-level variation), none of the three assemblers, including Matam, could reconstruct input sequences exactly. SPAdes and Emirge tend to collapse strain-level diversity into approximately species-level representative sequences, or they fail to produce an assembly in some cases. Because Matam tends to overpredict, SPAdes and Emirge were selected to be included in the phyloFlash pipeline. SPAdes was chosen as the default option, because it yields a compromise between the two extremes of chimerism and fragmentation. Overall, SPAdes assembles longer scaffolds that are potentially more divergent as it is not dependent on a reference database.

**SSU rRNA-based metagenome analysis with phyloFlash.** We designed phyloFlash (https://hrgv.github.io/phyloFlash/) for the rapid screening of SSU rRNA sequences in metagenomic libraries, through both a mapping-based overview of the taxonomic composition and full-length targeted assemblies. phyloFlash implements the database filtering described above and allows the user to choose between different mapping tools (SortMeRNA or BBmap) and assemblers (Emirge and/or SPAdes), to evaluate the various trade-offs discussed above in a consistent way. The results and summary statistics are formatted as standalone HTML files to make sharing and reporting easier. Additional utilities are also provided to compare the phyloFlash results from different libraries based on their taxonomic composition, e.g., graphically with bar plots or heatmaps or numerically as a distance matrix.

The default settings of phyloFlash have been chosen to allow quick run times on metagenomes of low to moderate diversity that have been sequenced as paired-end reads on Illumina platforms. High-diversity samples or those with uneven coverage, such as multiple displacement amplification libraries or metatranscriptomes, would likely need parameter optimizations.

Here, we present examples of how phyloFlash can be used to analyze a single metagenome, to compare multiple samples, and as part of a genome binning pipeline. Example commands to reproduce the phyloFlash runs are given in Data Set S2.

**(i) Low-diversity metagenome and reference database completeness.** We used a low-diversity, natural metagenome of known composition—a species of the marine catenulid flatworm *Paracatenula* sp. (32)—to evaluate the effect of reference database completeness on the taxonomic composition and targeted assembly reported by the phyloFlash pipeline. Each species from this flatworm genus has an obligate symbiosis with a corresponding species of intracellular bacteria from the candidate genus Riegeria (*Alphaproteobacteria*: *Rhodospirillaceae*), which makes up approximately 40% of the biomass (33). The sequence library is hence predominantly composed of two genomes, one eukaryotic and one prokaryotic.

The majority of reads recovered by phyloFlash were either classified as *Animalia* (79.5%) or *Rhodospirillales* (17.8%), based on their best mapping hits to the SILVA reference database (Fig. S6). Each of the two assemblers included in the phyloFlash pipeline, SPAdes and Emirge, assembled full-length SSU rRNA sequences of both target organisms. This was expected because SSU rRNA sequences with >99% identity to the targets were already available in the SILVA database.

When SSU rRNA sequences >87% identical to those of the target organisms were removed from the reference database, which is equivalent to removing relatives up to the family level (21), targeted assembly could still yield a consistent picture of the metagenome composition, but reference-based classification of the extracted reads was less reliable. SPAdes assembled the same two full-length sequences as before, whereas Emirge assembled only a partial animal 18S rRNA sequence (962 bp, of which 203 were Ns), and no bacterial sequences (Fig. S6). The read classification based on mapping still assigned the majority of SSU rRNA reads to *Animalia* (75.5%) but only 1.5% to *Rhodospirillales*. Instead there was a higher proportion assigned to unclassified *Alphaproteobacteria* (9.6%), unclassified *Proteobacteria* (3.2%), or other alphaproteobacterial taxa (5.0%) (Fig. S6).

In comparison, the assembler Matam, which aligns contigs to the SSU rRNA reference database during scaffolding, produced fragmented assemblies when relatives

were excluded from the reference database. Four sequences (508 to 740 bp) corresponded to the animal *Paracatenula* sp., and two (647 and 1,043 bp) to the bacterium *Ca*. Riegeria (Fig. S6). When the complete reference database was used, both the full-length target SSU rRNAs were assembled by Matam. This illustrates a possible drawback of reference-guided assembly approaches, such as those used by Emirge and Matam, in that their assemblies can be fragmented or incomplete when the target sequence is too divergent from existing reference sequences.

SSU rRNA-based metagenome profiling remains a powerful complement to read classifiers that use whole-genome data, e.g., Kaiju (13) and Kraken (14). This is because the reference databases for this gene are more phylogenetically representative than their equivalents for whole-genome data, particularly for eukaryotes, even though the SSU rRNA makes up only a small percentage of total read data in most metagenomes. When samples contain divergent taxa whose close relatives are not represented in the reference database, taxonomic profiles based on SSU rRNA read classification may be misleading or inconsistent, and targeted assemblers that depend on alignment to a reference database may perform poorly, as shown in this example. Misclassification of reads could be exacerbated by inconsistencies between the taxonomy and the actual gene phylogeny (34). However, if they are sufficiently well represented in the metagenome, full-length SSU rRNA sequences can still be assembled for divergent taxa by reference-independent assemblers, such as SPAdes. The targeted assembly approach has the additional benefit that assembled SSU rRNA sequences can be used for further phylogenetic analyses, which is not possible with taxonomic profiles, e.g., from read mapping.

**(ii) Comparison of multiple metagenome samples.** The taxonomic summary produced by the phyloFlash pipeline permits a rapid comparison of multiple metagenomes at a coarse-grained level, which can be followed by a finer-grained examination of specific taxa using the assembled sequences. To demonstrate the comparison of multiple samples, we applied the phyloFlash pipeline to metagenomes from six species of marine oligochaete worms of the genera *Olavius* and *Inanidrilus*. Each worm species in these genera has multiple species of symbiotic bacteria, with the most abundant symbiont being typically from the candidate genus Thiosymbion (*Gammaproteobacteria*: *Arenicellales*) (35). These worm species were chosen because their symbiont diversity is relatively low (no more than six bacteria per host species), and they have been characterized in previous studies by Sanger sequencing of SSU rRNA clone libraries and FISH (Data Set S1) (36–41). Publicly available sequencing data were directly downloaded and processed with phyloFlash using the script ENA_phyloFlash.pl that is provided with phyloFlash (Data Set S1).

The results files from phyloFlash were directly used by the phyloFlash_compare.pl script to plot a synoptic overview of the read mapping-based taxonomic composition for all samples, which can either be a heatmap (Fig. 6) or barplot. Certain bacterial taxa occurred in multiple animal species and hence were likely to represent secondary symbionts, e.g., *Desulfobacterales*, *Rhodospirillales*, and *Spirochaetales*. The samples also fell into two clusters, one where *Desulfobacterales* and *Spirochaetales* were both present, and the other where *Alphaproteobacteria* (especially *Rhodospirillales*) were more abundant.

**(iii) Connecting SSU rRNA sequences to metagenomic bins.** The SSU rRNA gene is an essential tool in microbial ecology, being used as a phylogenetic marker, and as a target for molecular probing to visualize specific taxa or species in environmental samples (19, 42). rRNA-targeted molecular ecology previously involved PCR amplification of the gene with specific primers, followed by one or more iterations of phylogenetic analysis, primer/probe design, microscopy, and PCR, the so-called rRNA cycle approach. Shotgun metagenomics can also yield full-length SSU rRNA sequences, assembled by phyloFlash or other tools, which can then be used as input into the "rRNA cycle." In addition, microbial genomes can be "binned" from metagenome assemblies (metagenome-assembled genomes, or MAGs), offering functional information not otherwise accessible through the classical "rRNA cycle."

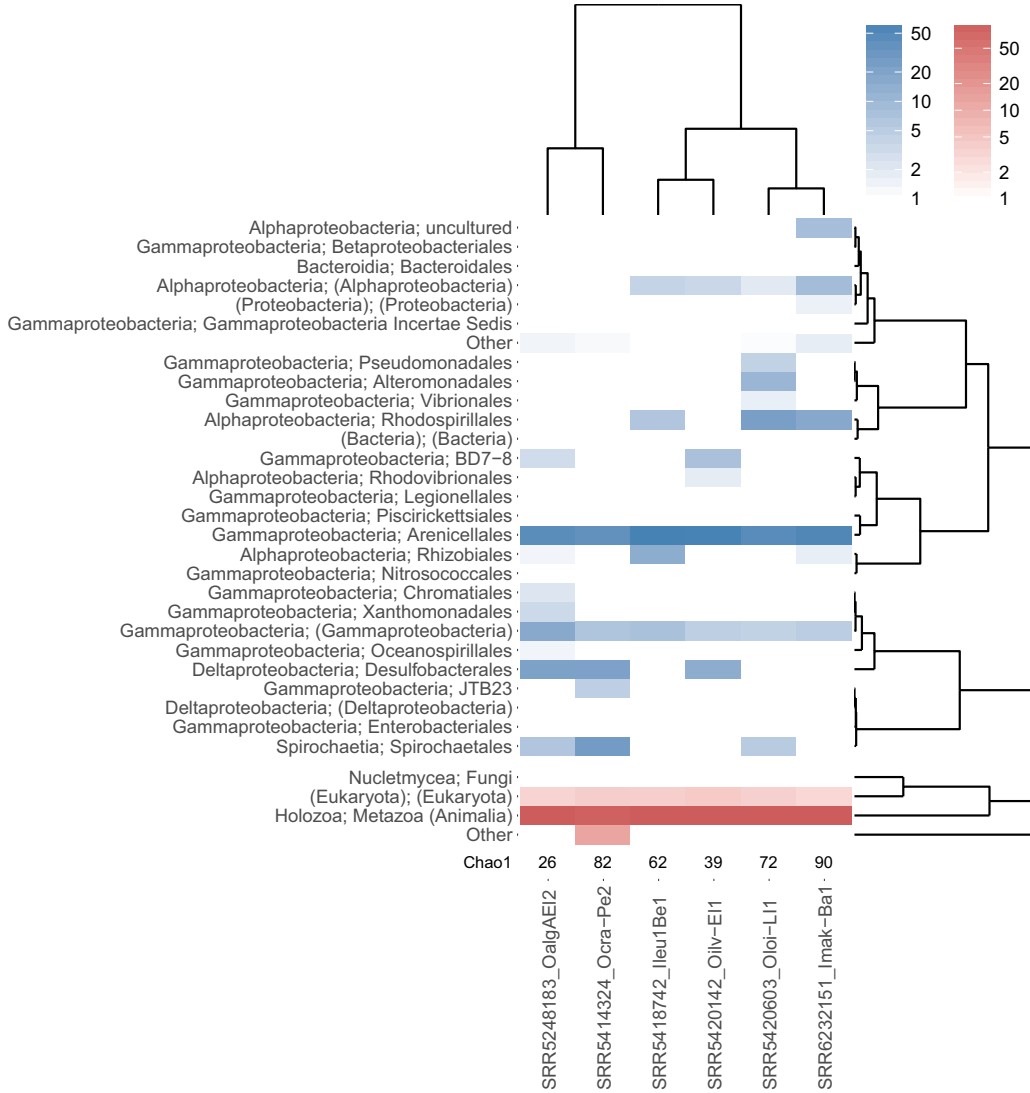

**FIG 6** Heatmap of taxonomic assignments (rows) for SSU rRNA reads in metagenome libraries of six gutless oligochaete species with symbiotic consortia (columns). The plot was generated with the comparison scripts provided with phyloFlash. Color intensities represent the percentage of reads mapping to a given taxon, separated by prokaryotes (blue) and eukaryotes (red).

However, MAGs often lack SSU rRNA sequences, as observed in several recent studies (1, 24). The SSU rRNA gene itself is difficult to assemble, because the gene has several highly conserved regions. These conserved regions lead to sets of shared k-mers among the SSU rRNA sequences present in the read pool that confound the assembly graph structure and eventually cause fragmented assemblies. Even if a full-length SSU rRNA gene is assembled, it may be difficult to assign to a specific MAG bin because the rRNAs often have multiple copies that result in the rRNA genes being assembled into separate contigs from the rest of the genome. In addition, binning strategies often cannot assign such rRNA contigs to their correct bins because the rRNA genes are compositionally different. Coverage-based binning approaches also have similar problems, again due to the higher and variable copy number of rRNA genes. Although MAGs without SSU rRNA genes can still be phylogenetically informative and lead to the discovery of new microbial diversity (43), the lack of an SSU rRNA means that these genomes cannot be linked to known clades that have been defined on the basis of SSU rRNA sequences (21), nor can they be targeted by molecular probing for visualization or sorting (44).

SSU rRNA sequences can be connected to MAGs using contig connectivity information contained in the assembly graph. Assemblers such as Megahit (45) and Meta-SPAdes (30) produce a graph-based representation of the assembly in the Fastg format (http://fastg.sourceforge.net/). This reports branching or circular connections between contigs, which are not retained when the assembly is resolved into linear scaffolds. Contigs connected to SSU rRNA sequences in the metagenome assembly graph can be "fished" with the script phyloFlash_fastgfishing.pl included in the phyloFlash software. This script identifies SSU rRNA sequences in the assembly, parses the assembly graph to find contigs connected to the SSU rRNA-containing contig, and optionally compares them to the SSU rRNA sequences from the initial targeted assembly from the main phyloFlash pipeline. This approach has been used to recover MAGs with SSU rRNAs from several host-associated metagenomes (46, 47).

In practice, which assembler and binning software produces the best results for a given data set has to be determined empirically. phyloFlash aids this process by giving a quick overview of the expected diversity in the library by SSU rRNA taxonomic profiling and targeted assembly and by facilitating the comparison of this initial overview to the complete metagenome assembly and derivative MAGs through connectivity "fishing" of SSU rRNA sequences.

**Conclusion.** The SSU rRNA gene remains indispensable to molecular ecology in the current era of whole-genome metagenomics, particularly because of the extensive reference data that have been accumulated for this gene, and the use of rRNA as a target for molecular probing. phyloFlash is a complete workflow for SSU rRNA-targeted profiling of metagenomes that integrates database preparation, read extraction, taxonomic classification, targeted assembly, and connecting SSU rRNA sequences to MAGs. It has been designed to provide a quick and user-friendly overview of the composition of metagenomic sequence libraries based on all domains of life, and it can be used for the rapid screening and comparison of multiple samples.

## MATERIALS AND METHODS

**Implementation of phyloFlash.** The phyloFlash pipeline is written in Perl 5, with additional scripts in Perl and R for downstream analyses such as visualizations or multisample comparisons. The pipeline is outlined in Fig. 1, and a manual is available at https://hrgv.github.io/phyloFlash/. The software is also distributed via the Bioconda channel of the Conda package manager.

**(i) Database filtering and preparation.** The pipeline uses the SSU Ref NR99 database from the SILVA project (https://www.arb-silva.de) (31). Sequences containing fragments of the large-subunit (LSU) rRNA are detected with a hidden Markov model (HMM) from a customized version of Barrnap (E-value cutoff of $10^{-10}$, >10% of total model length) (https://github.com/tseemann/barrnap) and are removed. Sequence regions containing low-complexity sequence or repeat k-mers are masked with bbmask.sh (from BBmap [https://sourceforge.net/projects/bbmap/]) using k-mer lengths between 4 and 8 to detect both repeats and low-complexity sequences, with a minimum masked sequence length of 20 bp, and entropy cutoff of 0.7. Known sequencing or cloning vectors are filtered by matching against the NCBI Univec database (ftp://ftp.ncbi.nlm.nih.gov/pub/UniVec/) with bbduk.sh (BBmap), using a k-mer length of 27 (down to 11 at the ends of the sequences), maximum Hamming distance of 1, and discarding sequences below 800 bp after trimming. The sequences are converted from RNA to DNA alphabet, and ambiguous bases are replaced by random base characters. The sequences are then clustered with VSEARCH cluster_fast at 99% ("NR99") and 96% identity ("NR96") (48). The NR99 sequences are indexed for BBmap with bbmap.sh and in the UDB format with VSEARCH (v2.5.0+). The NR96 sequences are indexed for Emirge (15) with Bowtie (49) and for SortMeRNA (29) with indexdb_rna (from SortMeRNA). The script phyloFlash_makedb.pl automates the database preparation steps described above. phyloFlash_makedb.pl can be used to update to new releases of the SILVA database and has been tested with the latest available SILVA release (132 at the time of submission).

**(ii) SSU rRNA read extraction.** The inputs for phyloFlash are shotgun metagenomic paired-end libraries, which have been generated by an Illumina sequencer. SSU rRNA reads can be extracted either with BBmap or SortMeRNA.

For BBmap, the input reads are aligned (mapped) against the filtered NR99 database, with a minimum identity of 70% by default, retaining all ambiguous alignments if there are multiple best-scoring hits. Insert size and mapping identity histograms are also reported to evaluate library quality. Output is written in SAM and Fastq formats, retaining all read pairs where at least one read could be aligned. Known limitations in the BBmap implementation of SAM bitwise flags and read name handling are addressed.

For SortMeRNA, the input reads are first reformatted to uncompressed interleaved Fastq format with reformat.sh (BBmap). The reads are then aligned against the filtered NR96 database, using a min_lis of 10 and E-value cutoff of $10^{-5}$, reporting the 10 best alignments. Aligned reads are reported in SAM and

Fastq formats. The SAM file is further processed to edit the bitwise flags, as SortMeRNA does not report complete pairing information in its implementation of the SAM format.

**(iii) Taxonomic summary from read mapping.** The edited SAM files from the read extraction step are read into memory. For each read pair, the taxonomic affiliation is assigned by taking the last common ancestor (LCA) of the taxonomy strings of all the database hits, using the SILVA taxonomy. The counts of LCA consensus taxa for the library are then summarized at a user-specified taxonomic level between domain and species (class by default).

**(iv) Assembly of full-length SSU rRNA sequences.** The extracted reads are assembled to full-length sequences using SPAdes (30) and optionally with Emirge.

For SPAdes, k-mer lengths are chosen based on the input read lengths. For reads of ≥134 bp, k-mers of length 99, 111, and 127 bp are used. For shorter reads, the k-mer lengths used are the next lower odd number to the read length minus 27, 17, and 7 bp. After assembly, contigs are screened for SSU rRNA sequences with HMM models for *Bacteria*, *Archaea*, and *Eukaryota* as implemented in the customized version of Barrnap. SSU rRNA regions under the E-value cutoff of $10^{-100}$ and at least 0.6 times the full-length model are extracted with Bedtools (50).

For Emirge, read pairs are run as single reads when their length is >152 bp. For paired-end input, the average insert size estimated from the initial mapping with BBmap is used if the insert size is greater than 2.2 times the read length; otherwise, a minimum insert size of 2.2 times the read length plus 0.5 is used. Emirge is run with 40 iterations.

To estimate the proportion of the extracted reads that were successfully assembled, the reads are remapped to the assembled sequences with bbmap.sh (BBmap) in fast mode using a minimum identity of 98%. The closest-matching database sequences are also found with usearch_global (51) as implemented in VSEARCH at a minimum identity of 70%. Assembled sequences and their closest database hits are then aligned with Mafft (52) to produce an alignment and guide tree.

**(v) Reporting of results.** Results are reported in both plain-text and HTML formats; the HTML report features interactive SVG-formatted graphics and allows the results to be shared easily in a single file that is compatible with modern web browsers without additional software. The report contains the taxonomic summary, mapping statistics, tree and tables of assembled full-length sequences and their closest database hits. All outputs, including mapping files and assembled sequences, can also be compressed into a single archive file at the end of the phyloFlash run.

**(vi) Comparison of multiple samples.** The phyloFlash results of multiple metagenomes can be graphically compared with the phyloFlash_compare.pl script. This uses the mapping-based taxonomic summary to generate either a barplot or heatmap of taxa for each library, at a user-specified taxonomic level. The clustering of libraries in the heatmap can either treat each taxon as independent or account for the taxonomy in calculating the distance matrix, using a Unifrac-like metric.

The abundance-weighted taxonomic Unifrac-like metric treats the hierarchical taxonomy as a tree where each taxon is represented by a node $i$ associated with a branch of unit length. The raw weighted Unifrac value $u$ of reference 53 therefore reduces to $u = \Sigma_i^n |\frac{A_i}{A_T} - \frac{B_i}{B_T}|$ where $n$ is the total number of nodes in the tree, $A_i$ and $B_i$ are the numbers of sequences in samples A and B, respectively, classified to node $i$ or its descendants, and $A_T$ and $B_T$ are the total numbers of sequences in samples A and B, respectively. Furthermore, as the taxonomy tree is ultrametric (all branches of the same length), the parameter $d_j$ of reference 53 reduces to the taxonomic level at which the comparison is performed, e.g., 4 for "order" in the Linnaean hierarchy. Then the scaling factor $D = 2d$ where $d$ is the taxonomic level, and the normalized abundance-weighted taxonomic Unifrac-like metric is $u' = \frac{u}{2d}$.

**Evaluation of unfiltered versus filtered database. (i) Database preparation and indexing.** The SILVA release 132 SSU Ref NR99 database was downloaded from arb-silva.de in Fasta format. For the unfiltered database, sequences were clustered at 96% identity with VSEARCH v2.5.0 ("NR96") and converted from RNA to DNA alphabet, and any ambiguous bases were replaced by random base characters. Indices were built for BBmap v37.99 and SortMeRNA v2.1b with default parameters. The filtered database was prepared as described above for the phyloFlash pipeline. Read mapping for this comparison was performed with an E-value cutoff of $10^{-5}$ for SortMeRNA (this is the default cutoff used by the Matam assembler) and minimum mapping identity of 70% for BBmap on 10 metagenomic libraries from the Tara Oceans project, using only the first 10 million read pairs per library (Data Set S1) (54).

**(ii) Calculation of sequence entropy and redundancy.** The number of possible DNA k-mers of length $k$ is $N = 4^k$. For a sequence of length $L$, k-mers are counted by applying a sliding window of width $k$ across the sequence. The total occurrences of a given k-mer $i$ is $w_i$. The total count is $\Sigma^N w_i = L - k + 1$. The probability of each k-mer is $p_i = w_i / \Sigma_i^N w_i$. Hence, the entropy of the sequence using k-mer length $H = -\Sigma_i^N p_i \log_2 p_i$.

The redundancy $R = 1 - H/\log_2 A$, where $A = N$ if $L - k + 1 > N$ and $A = L - k + 1$ otherwise. This is because if $L$ is short and $k$ is long, the maximum possible entropy of the sequence is constrained by the length of the sequence. Redundancy is thus always between 0 (maximum entropy) and 1 (zero entropy). The calculation was performed on the SAM file from each mapping with the script readEntropy.pl (https://github.com/kbseah/misc_tools/).

**Evaluation of read mapping and assembly methods. (i) Simulated metagenome "set 1."** Shotgun metagenome paired-end reads were simulated from 10 bacterial RefSeq genomes (Data Set S1) with the software package ART v2.5.8 (55) at average 50-fold coverage per genome, with the first library using the Illumina HiSeq2000 error profile (100-bp read length, insert size 220 ± 110 bp), and the second library using a HiSeq2500 profile (150-bp read length, insert size 300 ± 110 bp). Plasmid sequences were

excluded from the simulation. To compare the effect of different mapping settings on read extraction, the phyloFlash pipeline (v3.3) was run with either BBmap (at three different minimum identity cutoffs: 50%, 60%, and 70%) or with SortMeRNA (at three different E-value cutoffs: $10^{-5}$, $10^{-7}$, and $10^{-9}$) and using the filtered SILVA 132 SSU Ref database as described above ("Implementation of phyloFlash"). Overlap between the reads aligned by each method was calculated with the script samDiff.pl (https://github.com/kbseah/misc_tools/). Coverage of genomic features was calculated with featureCounts v1.5.2 using the RefSeq feature tables and the SAM-formatted alignment produced by ART to track the extracted reads back to the original genomes. Timings for BBmap and SortMeRNA read extraction given in Table S1 were based on runs on the same computer using eight processors (Intel Xeon CPU E5-1650 v2, 3.5 GHz).

**(ii) Simulated metagenomes "set 2" and "set 3."** Shotgun 100-bp and 150-bp paired-end reads were also simulated using ART, with the same settings as for "set 1" above, for two additional simulated metagenomes. "Set 2" comprised 25 genomes of *Bacteroides* spp. representing 25 strains in four species chosen from cultivated isolates of human gut microbiota (56). "Set 3" was a more diverse set that comprised 16 genomes of *Bacteroides* spp. representing 15 strains in 10 species, from the same source (Data Set S1).

**(iii) Targeted assembly from simulated metagenomes.** For each simulated metagenome, SPAdes v3.11.1 and Emirge v0.60 were used to assemble extracted reads within the phyloFlash pipeline, using the parameters described above. Each library was also separately assembled with Matam v1.4.0 (16) for comparison, using the filtered SILVA 132 SSU Ref NR96 database, extracting reads with SortMeRNA at an E-value of $10^{-5}$ and with other parameters at default. The scaffolds assembled by each tool from the read set extracted by SortMeRNA at an E-value of $10^{-5}$ were aligned against SSU rRNA sequences identified from the original genome assemblies by Barrnap, using Mafft v7.187 (FFT-NS-2 strategy) (52). The phylogeny was inferred from the alignment with FastTree v2.1.7 (57) using the Jukes-Cantor substitution model, and CAT approximation with 20 rate categories. Code and results for the simulated metagenome analyses are archived at https://doi.org/10.5281/zenodo.3909379 (set 1), https://doi.org/10.5281/zenodo.3909385 (set 2), and https://doi.org/10.5281/zenodo.4063152 (set 3).

**(iv) Environmental metagenomes.** A total of 85 metagenomic libraries from the Tara Oceans project (54) were used to evaluate performance on real-world data (Data Set S1). For each library, only the first 10 million read pairs were used to control for the effect of library size on assembly quality. Different mapping settings were tested as described above. Assembly with SPAdes, Emirge, and Matam was performed as described above. The uchime_ref chimerism score (5) was calculated for each assembled SSU rRNA gene sequence with VSEARCH v2.5.0, using the filtered SILVA 132 SSU Ref NR99 database as reference. Code and results for the environmental metagenomes analyses are archived at https://doi.org/10.5281/zenodo.3909388.

**Usage examples. (i) Low-diversity metagenome and reference database completeness.** The metagenomic library (BioProject accession no. PRJEB29979) of a small marine flatworm and its associated symbiotic bacterium was processed with phyloFlash with the option -everything (BBmap mapper, SPAdes and Emirge assemblers), using either the default SILVA 132 SSU Ref database or version of the database where all sequences with ≥87% identity to the flatworm and symbiont SSU rRNA genes were removed. Sequences were also assembled with Matam as described above for the simulated metagenome. Code and results from the low-diversity metagenome analysis are archived at https://doi.org/10.5281/zenodo.3909402.

**(ii) Comparison of multiple environmental metagenomes.** Publicly available metagenomic libraries from seven gutless oligochaete worm species, whose symbiotic bacteria had previously been characterized by Sanger-sequenced clone libraries of PCR-amplified SSU rRNA (Data Set S1), were processed with phyloFlash with the option -everything. Samples were compared as a heatmap on the basis of mapping-based taxonomic composition at the order level using phyloFlash_compare.pl with options -task heatmap,barplot -level 4. Code and results from the multiple metagenomes comparison are archived at https://doi.org/10.5281/zenodo.3909396.

**Data availability.** phyloFlash source code is available at https://github.com/HRGV/phyloFlash under a GNU GPL3 license, and it is also archived on Zenodo at https://doi.org/10.5281/zenodo.1327399. The software is also distributed via the Bioconda channel (https://bioconda.github.io) of the Conda package manager. The version described in this paper is v3.3.

## SUPPLEMENTAL MATERIAL

Supplemental material is available online only.

**FIG S1**, PDF file, 0.7 MB.
**FIG S2**, PDF file, 0.7 MB.
**FIG S3**, PDF file, 0.7 MB.
**FIG S4**, PDF file, 0.7 MB.
**FIG S5**, PDF file, 0.1 MB.
**FIG S6**, PDF file, 0.7 MB.
**TABLE S1**, DOCX file, 0.01 MB.
**TABLE S2**, DOCX file, 0.01 MB.
**DATA SET S1,** ODS file, 0.02 MB.
**DATA SET S2,** DOCX file, 0.007 MB.

## ACKNOWLEDGMENTS

This work was supported by the Max Planck Society (H.R.G.-V., B.K.B.S.) through Nicole Dubilier, and by Marie Curie Intra-European Fellowship (PIEF-GA-2011-301027 to H.R.G.-V.).

We thank our colleagues and phyloFlash users who have contributed to the software development by reporting bugs and suggesting new features. We also thank Nicole Dubilier for arranging financial support, Antonio Fernandez-Guerra for help with accessing test data, and the three anonymous reviewers for their suggestions for improving the manuscript. Oligochaete metagenome data in the usage examples were produced at the U.S. Department of Energy Joint Genome Institute in collaboration with the Max Planck Institute for Marine Microbiology under the Community Science Program.

H.R.G.-V. conceived the project and wrote the initial version of software. E.P. and B.K.B.S. developed software. B.K.B.S. and H.R.G.-V. tested software and analyzed data. B.K.B.S. and H.R.G.-V. wrote the manuscript.

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
