## [Reviewer comments · mSystems]

phyloFlash – Rapid SSU rRNA profiling and targeted assembly from metagenomes

Harald Gruber-Vodicka, Brandon Seah, and Elmar Pruesse

Corresponding Author(s): Harald Gruber-Vodicka, Max Planck Institute for Marine Microbiology

Review Timeline:

Submission Date:	September 14, 2020
Editorial Decision:	September 16, 2020
Revision Received:	October 2, 2020
Accepted:	October 2, 2020

Editor: Mani Arumugam

Reviewer(s): Disclosure of reviewer identity is with reference to reviewer comments included in decision letter(s). The following individuals involved in review of your submission have agreed to reveal their identity: Shingo Kato (Reviewer #1)

Transaction Report:

DOI: <https://doi.org/10.1128/mSystems.00920-20>

Response to reviewers

We thank the reviewers for their time and consideration in reviewing our manuscript (mSystems00206-20), and for their constructive suggestions.

Below, we first explain the major changes, before addressing each reviewer's comments point-by-point. Text in blue are the original reviews quoted verbatim; our response is in black. Line numbers quoted below correspond to the "Marked-Up Manuscript" version with visible tracked changes.

Major changes

1. Resubmitted to Methods and Protocols track

Reviewers 1 and 3 and the Editor stated that the article is a better fit for the Methods and Protocols track than the Research Article track, because it describes an improved method rather than novel research findings. We agree with this assessment and have resubmitted to the Methods and Protocols track.

2. Removed section on SSU rRNA genome "fishing"

Reviewers 1 and 3 both stated that the section "Assembly graph and SSU rRNA-based binning of microbial genomes" should be revised or removed because the "fishing" procedure and its novelty were not adequately described or supported. Specifically, the contigs recovered by "fishing" were not systematically compared with MAGs binned by other metagenome binning software, and the section read more like a tutorial than scientific results. Reviewer 1 also pointed out that the term "binning" is inappropriate for the contigs that are "fished" by graph connectivity.

We acknowledge the limitations of this section and have removed the results relating to "SSU rRNA-based binning" from the Results and Discussion (lines 555-627) and related sections in the Abstract (lines 22-24) and Materials and Methods (lines 249-263), as suggested. We retained the discussion of how metagenome binning software often do not recover the SSU rRNA gene along with the rest of the genome, and now simply refer to the assembly graph "fishing" procedure as a possible way to connect full-length SSU rRNA sequences to metagenomic bins produced by third party software, e.g. for quality control. We have removed any part that claimed that fishing alone was adequate to retrieve complete MAGs. The new heading for this section is "Connecting SSU rRNA sequences to metagenomic bins"

3. Compared assembler performance with simulated metagenome of closely related strains

Reviewer 3 explained the need to compare the performance of the different approaches for targeted assembly on more complex metagenomes containing closely related strains, because the simulated metagenome we used for testing contained only divergent species.

We agree on the need to evaluate closely related strains, which we expect to be more challenging to assemble. We prepared a second simulated metagenome comprising 25 strains (from 4 species) of *Bacteroides* isolates from human guts, and tested the three assemblers SPAdes, Emirge, and Matam with the same parameters as the first simulated metagenome (lines 205-208, 378-394).

Unexpectedly, none of the three assemblers could reconstruct any of the input sequences exactly. Similar to the performance on the divergent simulated metagenome, the reference-based Matam produced the most sequences, but the majority of these were short (< 60% full length SSU), and many also suffered from chimerism. SPAdes failed to assemble any sequence with the 100 bp PE library, and produced only two sequences with the 150 bp PE library. Emirge performed the best of the three, consistently producing three full-length sequences from each library (100 bp and 150 bp), of which two fell in species clusters and one was chimeric. The results with the second simulated metagenome are now reported at lines 378-394 and discussed in relation to the other simulated metagenome at lines 418-426.

We find that the poor performance of all three assemblers reflects inherent limitations of the information available in short reads to resolve strains. Resolving strain diversity in metagenomes is a developing field, and

single libraries and single loci are probably insufficient. Studies dealing with this problem take advantage of coverage variation across multiple samples and across whole genomes (e.g. <https://doi.org/10.1038/s41396-019-0475-z> and <https://doi.org/10.1038/s41564-019-0572-9>).

The assemblers implemented in the phyloFlash pipeline (SPAdes and Emirge) tend to “collapse” diversity into approximately species-level representative sequences (or fail to produce an assembly), which we find to be a more conservative result than the likely spurious and numerous sequences produced by Matam. Furthermore Emirge and Matam are also less successful at assembling divergent sequences when close relatives are not present in the database, as demonstrated with the low-diversity usage example where target sequences were removed from the reference database (lines 458-475).

Reviewer #1 (Comments for the Author):

Comments to mSystems00206-20

The authors developed and introduced a bioinformatic pipeline, called phyloFlash, which could be used for taxonomic profiling of 16S rRNA genes in Illumina shotgun metagenomic sequencing data. This pipeline included useful tools to reconstruct a full-length 16S rRNA gene sequence from the short reads using third party assemblers, to easily compare 16S-based community structures among metagenomes, and to connect 16S rRNA genes to contigs using a graph-based method. Usefulness of this pipeline was assessed using mock and real-world data. I agree with the significance and usefulness of this pipeline reported in this manuscript, and think it is worthy for publication. However, the section "Assembly graph and SSU rRNA-based...." (pages 15 to 18), should be totally revised or removed (please see the comments below).

We thank the reviewer for their constructive comments on the manuscript.

Major concerns:

The section "Assembly graph and SSU rRNA-based...." (pages 15 to 18), seems to serve as a short paper itself that includes Introduction (the first and second paragraphs), Methods (the third paragraphs), and Results and Discussion sections (the other paragraphs). The Methods and Results were not well described. The pipeline of phyloFlash appears just only to reconstruct a full-length 16S rRNA gene from short reads, and to connect the 16S rRNA gene at the end of a contig and another contig end based on assembly graph produced by third party assemblers. This is a sort of "fishing" as mentioned by the authors themselves (p16, L14?), not "binning". However, this section would be written as if phyloFlash could be used to perform "binning" i.e., to reconstruct MAGs. Therefore, this section, and related sentences in the other sections, should be totally re-written, or just removed.

In the paragraph starting with "To counter these..." (p16) more details of methods should be described. Did the authors use only the third party assemblers and phyloFlash_fastgfising.pl to perform genome binning? This could only produce a number of contigs, some of which would be connected with 16S rRNA gene, but not produce MAGs. Generally, a binning tool such as MetaBAT or CONCOCT, is needed to produce MAGs. In the paragraph starting with "SSU rRNA ..." (p16), more details of MAGs from the simulated metagenome should be shown, i.e., the values of completeness and contamination for each MAG, the numbers of contigs and protein-coding regions for each MAG, and comparison with the genomes of original microorganisms in the mock community, and more importantly, the numbers of MAGs containing 16S rRNA gene (and the numbers of this gene). Similarly, in the paragraphs starting with "We then ...", "In a next step ...", and "To test the screening..." (p16-17), more details of MAGs from the real-world samples should be shown, such as comparison with the previously reported MAGs, in addition to the information described above. In particular, honestly I could not believe that a complete closed genome could be obtained from a real-world sample only by the method described in the manuscript. Please describe details of the methods, and details of the complete genome obtained. The obtained genome sequence should be deposited into a public database, e.g.,

GenBank/EMBL/DDBJ, before publication.

We have removed the section “Assembly graph and SSU rRNA based...” as suggested, see “Major changes” above.

Minor concerns:

16S-based profiling of microbial community has a bias originated from a variety of the copy numbers of this gene per genome. In the Introduction section, this weakness of 16S-based profiling should be mentioned. Single-copy marker (e.g., ribosomal proteins of which genes are rarely horizontally transferred) gene-based profiling is better to show more precise community structure.

We have added the following sentence to the Introduction (lines 67-70):

“Another potential drawback of the SSU rRNA gene as a marker for molecular ecology is that a single genome may have multiple copies of the rRNA gene operon, in both eukaryotes and prokaryotes, so the abundance of rRNA sequences may not directly reflect cellular abundance in a community.”

P2, IMPORTANCE: In the sentence starting with "Many environmental bacteria", it is hard to understand what means the term "visualization". For "environmental bacteria", not only bacteria, but also archaea should be described.

We have changed “environmental bacteria” to “environmental microbes” (to also include archaea and eukaryotic microbes), and changed “visualization” to “visualization by molecular probes”. The sentence now reads:

“Many environmental microbes are only known from high throughput sequence data, but the SSU rRNA gene, the key to visualization by molecular probes and link to existing literature, is often missing from metagenome-assembled genomes (MAGs).”

P3, in the paragraph starting with "Ideally...", what is "the same problem"? please make it clear.

The sentence has been rewritten to:

“However, these should be considered together because they all involve the same target gene, and improvements to each task can directly lead to improvements in the others.”

P14, the paragraph starting with "The lack of ..." seems to be beyond the scope of this paper, and therefore could be removed. phyloFlash is a 16-based profiler with a large reference data that includes a variety of organisms because a 16S rRNA gene sequence is easily determined, but the Kaiju and Kraken are a whole-genome-based profiler with reference data containing only the genome-determined organisms. Furthermore, the authors used only one sample to compare these two profilers. Extremely biased. Thus, such comparison should be avoided.

This paragraph has been removed, as suggested. Instead we briefly mention the issue of database completeness and how there is a larger knowledge base available for the SSU rRNA gene, vs whole genome data for read classifiers. (lines 476-500)

P15, the paragraph starting with "The full-length sequences..." seems not to be informative, and could be removed. Indeed, the contents of this paragraph did not include "more specific information" provided from the full-length sequences. The authors mentioned about contamination in the previously reported metagenome. However, it is possible that the previous report did not just point it out due to less significance in that study even if noticed at that time. Therefore, such indication should be avoided.

This paragraph has been removed, as suggested (lines 520-530). We now briefly mention the advantage of targeted assembly vs. taxonomic profiling in the previous subsection, where it is more appropriate:

“The targeted assembly approach has the additional benefit that assembled SSU rRNA sequences can be used for further phylogenetic analyses, which is not possible with mapping-based taxonomic profiles.” (line 498-500)

P16, A submitted paper (Jaeckle et al.) should not be referred. Several papers regarding genome reduction for endosymbionts have already been published (e.g., McCutcheon JP, Moran NA. 2011. Extreme genome reduction in symbiotic bacteria. *Nat Rev Microbiol* 10:13-26.)

The section where this is referred to has been removed (see “Major changes”). The paper has now been published, and is also referred to elsewhere in the manuscript because of data from that study used for a usage example (reference 42, line 446).

Reviewer #2 (Comments for the Author):

This MS by Gruber-Vodicka*, Seah*, and Pruesse introduces the phyloFlash software for extracting full-length (or near full-length) SSU sequences from raw metagenomic data. It assembles SSUs not only from Bacteria, but also from Archaea and Eukaryotes. The pipeline is thus extremely useful for anyone dealing with massive Illumina datasets as a first species composition check. It is relatively easy to install with Conda (given that it relies on multiple dependencies), it is well-documented, it outputs interactive HTML reports, and its authors seem to frequently reply to issues raised at GitHub. It also comes with several great features such as the SSU baited 'binning' approach or visualization scripts.

We thank the reviewer for their constructive comments and suggestions for improving the pipeline.

I don't have any major issues with the program, please publish it. My comments are mostly minor suggestions on how to further improve the pipeline.

(1) Please implement automatic read length detection. Currently, this is the only option that prevents running the program in parallel for hundreds of libraries downloaded from various databases. For example, I can imagine running phyloFlash on all Illumina datasets available for a particular environment.

Automatic read length detection has now been implemented and will be available in the next release (v3.4).

(2) My experience with EMIRGE is very similar to what is reported by this manuscript -- it tends to assemble numerous chimeric sequences. I see no real benefit in keeping it in the main pipeline. I would consider removing it at least from the -everything option.

The -almosteverything option runs everything except Emirge; we implemented this for the same reasons as the reviewer has described above. The ‘everything’ option retains Emirge for completeness’ sake.

(3) I understand that adding custom databases automatically is likely really tricky to implement. However, it would be really useful to provide guidelines on how to create custom databases (from curated SSU alignments or any other gene of interest such as LSU). One database that would be amazing to have for eukaryotes is the Protist Ribosomal Reference database (PR2) because it contains a field with SSU origin (nucleus, plastid, mitochondrion, nucleomorph).

We agree that better support for custom databases would be desirable. We have also lately implemented the option to adjust minimum reference sequence length for custom databases, in response to user feedback.

Instructions on formatting custom databases are documented in the online manual:

<https://hrgv.github.io/phyloFlash/install.html>

However we have found that only a small number of users have used custom databases for phyloFlash. The PR2 database is an attractive alternative for protists in particular, but the phyloFlash pipeline is at the moment closely tied to the SILVA taxonomy framework, which we chose because it covers all domains of life.

(4) The phylogenetic inference is not easy to customize. For example, I would appreciate at least: (i) allowing users to switch from NJ to ML; (ii) adding more than one closest sequence from the database (useful for highly divergent taxa).

We acknowledge that the tree is not easily customized. The tree produced in the pipeline is the guide tree from the Mafft aligner, and was intended primarily for visualization of sequence distances within the context of the graphical report. We decided against implementing more customization of the tree building procedure within the phyloFlash pipeline, because we expected that “advanced” users who wish to have more sophisticated phylogenetic analyses would want to use their own preferred software and would not accept the default output as-is.

(5) Since the SPAdes assembler accepts Sanger, PacBio, and Oxford Nanopore reads for hybrid assemblies with Illumina, could long reads be provided only for the assembly step or is there any reason not to do so?

There is the option to supply “trusted contigs” to compare against the assembled SSU rRNA sequences, e.g. sequences from reference genomes. However if raw, unassembled long reads were to be used, those reads containing SSU rRNA sequences would first have to be filtered out, which would essentially mean a reimplementing of the phyloFlash pipeline for long reads. Given the different error profiles and characteristics of long read sequencing, we decided that this would be beyond the current scope of the phyloFlash pipeline, but will keep it in mind for a future version.

Reviewer #3 (Comments for the Author):

Gruber-Vodicka et al present phyloFlash, a tool designed to analyze shotgun metagenomes by looking at the SSU rRNA reads. They show that cleaning up SSU rRNA databases has a major impact on the accuracies of taxonomic annotation of metagenomes. They also compare two methods to identify short reads coming from SSU rRNA genes and show their strengths and weaknesses. They further show that general purpose metagenome assemblers perform better than reference-based assemblers for obtaining full-length SSU rRNA sequences. Finally, they also show how "fishing" for SSU rRNA gene sequences in assembly graphs can improve completeness of metagenomically assembled genomes and their taxonomic annotation.

Some of the conclusions made in this manuscript are relevant for the field - e.g., need to clean up SSU rRNA databases, and the more surprising result that general-purpose metagenome assemblers did better than reference-based assemblers designed to assemble full length SSU rRNA sequences. However, they have failed to deliver on the promise of improving/completing MAGs by adding the SSU rRNA sequences, which would be quite a novel contribution (see below for detailed comments). Without this, the manuscript lacks novelty and is a comparison of different software programs for optimizing a pipeline, which should be then considered a protocol paper.

We thank the reviewer for their constructive comments on the manuscript.

The manuscript has been resubmitted to the Methods and Protocols track, as described in “Major changes” above.

Major concerns:

1. Simulated metagenomes for evaluation:

The authors simulated metagenomes consisting 10 genomes. This is a low to at-best low-medium complexity metagenome, with members that are quite distant from each other. How will their pipeline perform in recovering full length SSU rRNA sequences if multiple related species are present in the same community - e.g., like the multiple *Bacteroides* species present together in the human gut? How will the evaluations look? Will the general-purpose assembler still be better than the reference-based assemblers that can take hints from databases? This is a prerequisite for what the authors claim can be a disadvantage-turned-to-advantage of fishing for specific organisms.

We address this point under “Major changes” above.

2. SSU rRNA-based binning:

While the article is listed as a Research Article, the presentation of the article focuses more on the software and its capabilities rather than the novelty they bring. The entire section "SSU rRNA-based metagenome analysis with phyloFlash" reads like a software manual than a results section of a research article. Nevertheless, this is the only part that is relevant to the title.

We are unsure what is meant by “this is the only part that is relevant to the title”. The “targeted assembly” in the title refers to the targeted assembly of the SSU rRNA gene, which is addressed in several sections, primarily in “General-purpose assembler SPAdes yields ...”.

In the abstract, the authors motivate this work by the missing SSU rRNA genes inside MAGs, which I agree is a very important gap that need to be filled. In the text though, they have hidden it way inside a software tutorial and do not explain the details behind it (they only name the script that does this), nor do they perform a thorough assessment of the improvements they offer over current MAG approaches. There are several unanswered questions related to this part:

- (i) Do they bin assembled contigs using an external software and then connect bins to full length SSU sequences, or do they simultaneously bin assembled contigs and connect them to SSU sequences using their own script?
- (ii) How does their binning procedure compare to other binning programs such as MetaBAT2 and VAMB? If the same full length SSU sequences and assembled contigs are be provided to other binning programs, will they do better or worse than phyloFlash? Without an answer to this question, it is not clear whether there is any new contribution from this work with regards to binning and MAGs. This evaluation must also be done in complex real/simulated microbiome samples to assess performance in realistic scenarios.

We acknowledge the limitations of the section on MAG retrieval by SSU rRNA “fishing”, which were also raised by Reviewer #1. We have removed the results presented here, as described under “Major changes” above, and refocused the manuscript on the SSU rRNA -based profiling and SSU targeted assembly/reconstruction.

Minor concerns:

The supplementary material is a whopping 36.4 GB at zenodo website. I didn't bother downloading it given this size. The authors should make them available as individual components so that users can download only what they need rather than a huge tarball.

We have broken up the supplement into individual parts and reference them in the respective subsections of the Materials and Methods (lines 219, 227, 241, 248).

September 16, 2020

Dr. Harald R Gruber-Vodicka
Max Planck Institute for Marine Microbiology
Bremen
Germany

Re: mSystems00920-20 (phyloFlash - Rapid SSU rRNA profiling and targeted assembly from metagenomes)

Dear Dr. Harald R Gruber-Vodicka:

I have received reviews of your resubmitted manuscript from the same 3 reviewers who reviewed the original submission. They are quite positive about the revised manuscript. Before accepting the manuscript, I would like you to address the comment from Reviewer #3 - he might be right that you have been too hard on yourselves with the newly added simulated dataset "set 2", and that a more realistic combination of close species could evaluate the approach more realistically.

Below you will find the comments of the reviewers.

To submit your modified manuscript, log onto the eJP submission site at <https://msystems.msubmit.net/cgi-bin/main.plex>. If you cannot remember your password, click the "Can't remember your password?" link and follow the instructions on the screen. Go to Author Tasks and click the appropriate manuscript title to begin the resubmission process. The information that you entered when you first submitted the paper will be displayed. Please update the information as necessary. Provide (1) point-by-point responses to the issues raised by the reviewers as file type "Response to Reviewers," not in your cover letter, and (2) a PDF file that indicates the changes from the original submission (by highlighting or underlining the changes) as file type "Marked Up Manuscript - For Review Only."

Due to the SARS-CoV-2 pandemic, our typical 60 day deadline for revisions will not be applied. I hope that you will be able to submit a revised manuscript soon, but want to reassure you that the journal will be flexible in terms of timing, particularly if experimental revisions are needed. When you are ready to resubmit, please know that our staff and Editors are working remotely and handling submissions without delay. If you do not wish to modify the manuscript and prefer to submit it to another journal, please notify me of your decision immediately so that the manuscript may be formally withdrawn from consideration by mSystems.

Sincerely,

Mani Arumugam

Editor, mSystems

Journals Department
Reviewer comments:

Reviewer #1 (Comments for the Author):

The revised manuscript has been greatly improved. I have no more comments.

Reviewer #2 (Comments for the Author):

The authors have addressed all my questions and I'm happy with their answers. I have no further comments.

Reviewer #3 (Comments for the Author):

The manuscript is much improved and much cleaner.

I commend the authors on adding another simulated metagenomic dataset. To be honest, they made it quite hard for themselves by simulating a very difficult microbiome where 18 strains from the same species were present. In a typical gut microbiome these many strains from the same species may not be equally abundant (I am assuming they are equally abundant based on the information in Line 192 in Methods). Their results could turn gut microbiome researchers away from using this tool, because only a couple of SSU genes could be assembled from 25 genomes. But that judgement would be unfair to the method, because the devil is in the details and the authors have been too tough on themselves. As it stands now, we can conclude that (i) communities with distinct species can be assembled very well, and (ii) communities with many close strains will most likely not assemble very well. Point (i) is promising, and point (ii) is gloomy. My suggestion would be to simulate a mix of 10-15 Bacteroides species, where some of them (not all) have 2-3 strains, and process with phyloFlash. This is a simpler (compared to set2) but a more realistic problem to solve. And I have feeling that they would retrieve more complete SSU genes, which would be more encouraging to the gut microbiome community.

One minor comment: Please clarify how many species the simulated set2 has. It says 5 species in

Line 379 and 4 species in Line 207. The Table lists two *Bacteroides* sp. - are they close enough to be considered the same species?

Comments to mSystems00920-20

The revised manuscript has been greatly improved. Great job! I have no more comments.

Response to reviewers mSystems00920-20

We thank the three reviewers and the editor for their time and work in reviewing our manuscript. We appreciate the encouraging comments and suggestions for improvement that have been made.

The line numbers below correspond to the manuscript version with tracked changes.

Reviewer 3 requested a more realistic mixture of strains/species for the simulated metagenome example, as the existing example "set 2" was possibly too stringent:

> I commend the authors on adding another simulated metagenomic dataset. To be
> honest, they made it quite hard for themselves by simulating a very difficult
> microbiome where 18 strains from the same species were present. In a typical
> gut microbiome these many strains from the same species may not be equally
> abundant (I am assuming they are equally abundant based on the information in
> Line 192 in Methods). Their results could turn gut microbiome researchers away
> from using this tool, because only a couple of SSU genes could be assembled
> from 25 genomes. But that judgement would be unfair to the method, because the
> devil is in the details and the authors have been too tough on themselves. As
> it stands now, we can conclude that (i) communities with distinct species can
> be assembled very well, and (ii) communities with many close strains will most
> likely not assemble very well. Point (i) is promising, and point (ii) is
> gloomy. My suggestion would be to simulate a mix of 10-15 Bacteroides species,
> where some of them (not all) have 2-3 strains, and process with phyloFlash.
> This is a simpler (compared to set2) but a more realistic problem to solve.
> And I have feeling that they would retrieve more complete SSU genes, which
> would be more encouraging to the gut microbiome community.

As suggested, we have simulated a third metagenome, "set 3", comprising 16 genomes from 10 species, where 5 species are represented by 2 or 3 strains each. In line with the prediction above, more complete SSU rRNA genes were retrieved, though the same limitations that we uncovered were apparent: all reconstruction/assembly tools yielded at least one chimeric sequence (though Matam produced the most chimerism and fragmented assemblies by far), and close strains do not assemble well with any tool.

This is now reported in lines 198-203, 354-380 and in updated Supplementary File 1 and the additional panel B to Supplementary Figure 5.

Interestingly, we also detected SSU rRNA genes from unrelated organisms in this data set, which probably come from errors in metagenome binning. This underscores the importance of SSU rRNA marker-gene targeted analyses even for genomic data.

> One minor comment: Please clarify how many species the simulated set2 has. Is
> says 5 species in Line 379 and 4 species in Line 207. The Table lists two
> Bacteroides sp. - are they close enough to be considered the same species?

Thank you for catching this. "Set 2" contains 4 species, as the strains named AM10-21B and OM08-17BH have over 99.9% SSU rRNA sequence identity. This has been corrected in the text at line 355.

In addition to the changes requested above, we have also updated Table 1 with the latest information about the software tools listed there.

October 2, 2020

Dr. Harald R Gruber-Vodicka
Max Planck Institute for Marine Microbiology
Bremen
Germany

Re: mSystems00920-20R1 (phyloFlash - Rapid SSU rRNA profiling and targeted assembly from metagenomes)

Dear Dr. Harald R Gruber-Vodicka:

Your manuscript has been accepted, and I am forwarding it to the ASM Journals Department for publication.

Please note that Supplementary file 1 is missing the details of simulated metagenome "set 3". If the production editor does not ask for this, I request you to voluntarily add this information during the next available opportunity.

For your reference, ASM Journals' address is given below. Before it can be scheduled for publication, your manuscript will be checked by the mSystems senior production editor, Ellie Ghatineh, to make sure that all elements meet the technical requirements for publication. She will contact you if anything needs to be revised before copyediting and production can begin. Otherwise, you will be notified when your proofs are ready to be viewed.

Sincerely,

Mani Arumugam
Editor, mSystems

Journals Department
Supplemental Figure 5: Accept
Supplemental Figure 3: Accept
Supplemental Table 2: Accept
Supplemental Figure 6: Accept
Supplemental Figure 2: Accept
Supplemental Table 1: Accept
Supplemental Figure 1: Accept
Supplementary Data: Accept
Supplemental Figure 4: Accept